# Current Therapeutical Approaches Targeting Lipid Metabolism in NAFLD

**DOI:** 10.3390/ijms241612748

**Published:** 2023-08-13

**Authors:** Manuela Vitulo, Elisa Gnodi, Giulia Rosini, Raffaella Meneveri, Roberto Giovannoni, Donatella Barisani

**Affiliations:** 1School of Medicine and Surgery, University of Milano-Bicocca, 20900 Monza, Italy; m.vitulo1@campus.unimib.it (M.V.); elisa.gnodi@unimib.it (E.G.); raffaella.meneveri@unimib.it (R.M.); 2Department of Biology, University of Pisa, 56021 Pisa, Italy; giulia.rosini@student.unisi.it (G.R.); roberto.giovannoni@unipi.it (R.G.)

**Keywords:** nonalcoholic fatty liver disease (NAFLD), nonalcoholic steatohepatitis (NASH), cirrhosis, peroxisome proliferator-activated receptor (PPAR), farnesoid X receptor (FXR), natural compounds

## Abstract

Nonalcoholic fatty liver disease (NAFLD, including nonalcoholic fatty liver (NAFL) and nonalcoholic steatohepatitis (NASH)) is a high-prevalence disorder, affecting about 1 billion people, which can evolve to more severe conditions like cirrhosis or hepatocellular carcinoma. NAFLD is often concomitant with conditions of the metabolic syndrome, such as central obesity and insulin-resistance, but a specific drug able to revert NAFL and prevent its evolution towards NASH is still lacking. With the liver being a key organ in metabolic processes, the potential therapeutic strategies are many, and range from directly targeting the lipid metabolism to the prevention of tissue inflammation. However, side effects have been reported for the drugs tested up to now. In this review, different approaches to the treatment of NAFLD are presented, including newer therapies and ongoing clinical trials. Particular focus is placed on the reverse cholesterol transport system and on the agonists for nuclear factors like PPAR and FXR, but also drugs initially developed for other conditions such as incretins and thyromimetics along with validated natural compounds that have anti-inflammatory potential. This work provides an overview of the different therapeutic strategies currently being tested for NAFLD, other than, or along with, the recommendation of weight loss.

## 1. Introduction

The liver plays a pivotal role in the metabolism of lipids and lipoproteins [1], and metabolic disfunction, as well as altered hepatic signalling, can lead to the development of nonalcoholic fatty liver disease (NAFLD), which includes nonalcoholic fatty liver (NAFL) and nonalcoholic steatohepatitis (NASH) [2,3]. NAFLD is a chronic liver disorder characterized by lipid overload or steatosis. Donnelly et al. analysed the specific contribution of each route of hepatic fat accumulation, showing that it can result from increased lipolysis of adipose tissue (59%), de novo lipogenesis (26%) and diet (15%) [4]. In addition, the inhibition of fatty acid (FA) oxidation can reduce the secretion of FA and increase their absorption, thus contributing to the development of NAFLD [4,5]. Particularly, dysregulation both in the synthesis and in the β-oxidation of free fatty acids (FFAs) can worsen the NAFLD condition, as reported by Sunny et al., who showed an important association between high fat intake and the impairment of lipid metabolism [6]. NAFL can progress to NASH, as explained by the so-called “multiple hit” hypothesis, which recapitulates all the inflammatory events triggered by liver lipid accumulation, including reactive oxygen species (ROS) production, endoplasmic reticulum (ER) stress, and hepatocyte apoptosis [7,8]. Moreover, NASH can lead to the deposition of extracellular matrix, alteration to hepatic architecture, fibrosis, and cirrhosis, with a higher risk of developing hepatocellular carcinoma (HCC). To date, NASH is the leading indication for liver transplantation in the USA [9]. It has been estimated that the prevalence of NAFLD has increased from 25% to 38% over the last three decades together with increasing obesity, and that one third of the world’s population is affected by this disease [10]. The metabolic syndrome is characterised by several risk factors that increase an individual’s likelihood of developing atherosclerosis, cardiovascular disease (CVD) and type 2 diabetes mellitus (T2DM). However, NAFLD and the metabolic syndrome share a common pathophysiology, with insulin resistance as the key factor. Furthermore, the main cause of death in these patients is cardiovascular events [10]. The management of NAFLD, in accordance with the European Association for the Study of the Liver (EASL) guidelines [11], should rely on lifestyle changes, including diet and daily physical activity. However, many patients are unable to achieve or maintain a significant weight loss, and the absence of an approved pharmacological treatment for NAFL/NASH has prompted researchers to evaluate new potential therapeutic targets. This review focuses on the role of the lipid transport system and of lipoproteins within the liver in NAFLD; secondly, the most relevant therapeutic approaches in preclinical as well as in clinical studies are discussed (as shown in Figure 1).

### 1.1. Free Fatty Acids (FFAs) Role

Triglycerides ingested through diet are emulsified in the small intestine by bile salts and hydrolysed to FAs and 2-monoacylglycerol (2-MAG) by pancreatic lipase [12]. The absorption by the intestinal epithelium occurs mainly through simple diffusion, although some proteins, such as fatty acid transport protein 4 (FATP4) can facilitate the passage of long-chain fatty acids. On the contrary, intestinal absorption of cholesterol occurs through the transporter Niemann–Pick C1-like 1 (NPC1L1) and scavenger receptor class B, type I (SR-BI). In the enterocytes, FAs and 2-MAG are reassembled into triglycerides and, with cholesterol, transported into the endoplasmic reticulum where they form—in the presence of the protein Apo-B 48—chylomicrons that enter the lymphatic system and bloodstream via facilitated transport [13]. The uptake of FAs by different tissues varies according to their concentration within the cells and can occur by diffusion or due to specific transporters such as CD36 and FATPs [14]. The main translocation process across the plasma membrane involves plasma membrane FA-binding proteins (FABPpm), members of the FA transport protein (FATP) family, with FATP2 and FATP5 isoforms being expressed in the liver. The rate of cellular FA absorption is also driven by the presence of CD36 on the cell surface, which is regulated by the subcellular vesicular recycling of CD36 from endosomes to the plasma membrane [15,16]. In this regard, the main ligands of CD36 are lipoproteins like VLDL, LDL, and HDL [14]. On the inner side of the membrane, FAs move to the aqueous phase to bind to the cytoplasmic FABP [17]; even in this case, the CD36 receptor can play a role, facilitating the transport and providing a docking site for FABP or enzymes that act on fatty acids, such as acyl-CoA synthetase [18].

In hepatocytes, hepatic lipase (HL) produces FFAs from the lipoproteins and TGs that enter the cells through facilitated transport and diffusion [19]; FFAs can be β-oxidized to produce energy, esterified to TGs used for the production of very low density lipoproteins (VLDLs) [20] or stored in lipid droplets for later use.

Lipid transport, however, is also related to another phenomenon part of the metabolic syndrome, i.e., insulin resistance (IR). Systemic insulin resistance consists of impaired insulin-mediated suppression of hepatic glucose production. Adipose tissue IR refers to the loss of lipolysis suppression when insulin levels increase, and this is also associated with higher levels of circulating FFAs and increased hepatic gluconeogenesis [21]. This could lead to an increased uptake and subsequent accumulation of FAs in the liver, thus resulting in liver steatosis. In turn, this could further increase adipose tissue dysfunction, which, together with insulin resistance, results in the accumulation of lipid species such as diacylglycerol and ceramides [22]. In this regard, a work by Kalavalapalli et al. [23] investigated the possible differences between adipose tissue and liver tissue IR in diabetic patients with NASH, finding that adipose tissue IR correlated with the severity of liver fibrosis. On the contrary, hepatic IR is less affected, and considered a consequence rather than the cause of injury [24]. As previously reported, there are several types of bioactive lipid species involved in lipotoxicity, including ceramides. The latter are synthetized both in the liver and intestine and can directly act on mitochondrial β-oxidation, thus promoting the accumulation of triglycerides and the production of ROS. In addition, ceramides also promote the apoptotic process by activating the BAX complex within the mitochondria. At the intestinal level, ileum and cecum produce ceramides through the activation of the FXR pathway, whereas, conversely, bile acids such as cholic or chenodeoxycholic acids inhibit their synthesis [25].

The concentration of FFAs in the liver is directly related to the influx of FAs, suggesting a concentration-dependent mechanism, in particular for short- or medium-chain FAs [26]. The type of FAs also seems relevant in the induction of hepatocyte damage; in a work by Li et al., the authors demonstrated that the treatment of murine or human hepatocytes with monounsaturated FAs induced lipid overload without changes in viability, whereas the incubation with saturated FAs caused not only lipid accumulation but, above all, an increase in cellular apoptosis [27]. Due to the important physiological role of fatty acid transporters, alteration in their expression or function can play a role in the pathogenesis of disorders such as NAFL/NASH. Wilson et al. showed that *CD36* knockout mice fed with Western diet were protected from liver steatosis and insulin resistance, supporting the idea that enhanced FA uptake is an essential point in the progression of hepatic TG overload [28]. Similar results were obtained by Falcon et al., who reported that mice lacking FATP2 or FATP5 had a reduction in liver FA uptake, even in the presence of high-fat feeding. Not only the uptake of FAs, but also their degradation can determine the amount of intracellular lipids [29]. In fact, lipid autophagy has a beneficial role in hepatic homeostasis since it participates in the physiological lipid turnover, degrading the accumulated lipid droplets. Interestingly, there are interactions between CD36 and the intracellular processing of FAs, since the overexpression of CD36 results not only in increased FA absorption, but also in decreased β-oxidation and autophagy, contributing to the development of fatty liver disease [30,31]. Although these data have been obtained in animal models, CD36 has been detected upregulated in NAFL/NASH patients as compared to healthy controls, suggesting a possible role in prompting lipid accumulation [32,33]. Moreover, a clinical study on NAFLD patients showed that the expression of FATP5 was inversely associated with the histological findings of liver steatosis [18]. In the liver, cholesterol (together with triglycerides) is incorporated into VLDLs, which are secreted into plasma, where triglycerides are again broken down by lipoprotein lipase. By the loss of triglycerides, VLDL particles are, through a complex series of reactions, converted into LDL particles that are triglyceride-poor but cholesterol-rich. Via LDLs, cholesterol is delivered to and deposited in peripheral tissues, entering the cells using LDL receptors. The liver also plays an important role in the addition of cholesterol to circulating particles through the efflux of cholesterol from hepatocytes and macrophages. This process, defined as reverse cholesterol transport (RCT), employs specific transporters, such as ATP-binding cassette transporter A1 (ABCA1) and ATP-binding cassette subfamily G member 1 (ABCG1). ABCA1 adds cholesterol and phospholipids to the lipid-free APOA-I, thus resulting in a pre-βHDL, whereas ABCG1 induces the addition of cholesterol and phospholipids to the pre-β HDL, forming larger HDL particles or α-HDL [34,35]. Liver expression of the ABCA1 gene is localized in hepatocytes and resident macrophages, whereas the ABCG1 gene is mainly expressed in macrophages [36]. Preclinical and clinical data suggest that any dysfunction in this system can lead to the development of inflammatory diseases like atherosclerosis and NASH, since a reduced efficiency of these efflux systems can lead to intracellular lipid accumulation [34,35,37].

The expression of transporters involved in the uptake of FAs, their metabolism, and efflux mechanisms are regulated by a series of events, starting from epigenetic mechanisms to posttranscriptional ones. Among them, an important role is played by different transcriptional factors, namely those belonging to the peroxisome proliferator-activated receptor (PPAR) family, retinoid X receptor (RXR), liver X receptor (LXR), and farnesoid X receptor (FXR).

### 1.2. Nuclear Receptors

#### 1.2.1. Peroxisome Proliferator-Activated Receptors (PPARs)

PPARs are a family of transcription factors activated by FA metabolites that regulate the transcription of a large variety of genes, including those involved in lipid and cholesterol metabolism. Among them, PPARα is expressed in cells that exhibit high levels of β-oxidation and mitochondrial activity, such as hepatic and cardiac tissues. As shown in Figure 2, PPARs can directly regulate the transcription of target genes since they form a heterodimer with the retinoid receptor X (RXR) and bind to the PPAR response element (PPRE) sequences in the promoter of target genes via DNA-binding domains [38]. FAs, especially long-chain fatty acids, are natural ligands of PPARs [39], but various metabolites of FAs, such as acyl-CoA esters or oxidized FAs, can also activate these transcription factors [14]. Among others, PPARs are able to induce the expression of the CD36 transporter, regulating FA hepatocyte uptake [39], although the regulation of this gene can involve other organs as well. In this regard, in a preclinical work [40], after the complete ablation of the *Cd36* gene in *ob/ob* mice, the animals developed severe hepatic steatosis. Conversely, a more recent study by Zeng et al. [41] showed that the hepatic deletion of the *Cd36* gene (*Cd36LKO*) in mice under a high fat diet regimen improved insulin resistance and fatty liver. Taken together, these findings could suggest that even if PPARs are able to induce the expression of the *Cd36* gene, their action is not strictly related to the liver tissue but involves a response at the systemic level. PPARα activity is protective against liver steatosis, as demonstrated in animal models; in a rat model of NAFLD, its expression was significantly downregulated compared to that of normal rats [42], and Montagner et al. showed that the hepatic deletion of PPARα altered liver FA homeostasis and led to the development of hepatic steatosis/steatohepatitis [43]. These results were confirmed in a clinical study [44], where PPARα was found to be downregulated in the liver of NAFLD patients; thus, PPARα agonists can be regarded as possible molecules for the treatment of NAFLD. On the other hand, PPARγ induces an increase in the accumulation of FAs within fat cells, promotes the transition of macrophages from M1 to M2 phenotype [45], and is capable of regulating reverse cholesterol transport, prompting ABCA1 gene expression [46]. In addition, PPARγ can induce the expression of CD36, thus increasing the transport of FAs into hepatocytes [47]. The data obtained in animal models, as well as in humans, regarding the role of PPARs in NAFL/NASH are in fact contradictory. Zhang et al. reported that, in a mouse model, the activation of PPARγ was able to improve NASH through the inhibition of miR-21-5p [48], whereas Ni et at. showed that loss of PPARγ in macrophages was causing NASH progression [49]. Conversely, other studies demonstrated the importance of the role played by PPARγ, since its activation was essential for hepatic lipid accumulation and NASH development [50,51,52,53]. In humans, an increase in PPARγ mRNA expression levels was detected in the liver of obese patients with NAFLD who underwent bariatric surgery [47,54], and this finding was also associated with an increased expression of its downstream targets, including CD36 [47].

#### 1.2.2. Liver X Receptors (LXRs)

LXR transcription factors are involved in liver lipogenesis and are important for FA, cholesterol, and glucose homeostasis. These nuclear receptors are considered sensors of sterol elements, and they protect cells from lipid overload through the activation of several genes implicated in RCT, i.e., *ABCA1* and *ABCG1*. Indeed, cell cholesterol overload promotes the formation of oxysterols, leading to the activation of LXR and subsequent upregulation of ABCA1, as shown by Ghoneim et al. in rats fed with a high-fat diet, in which an increase in the expression of LXRα and LXRβ, as well as of their target genes *ABCA1* and *ABCG5*, was detected [55]. In NAFLD patients, it was shown that the expression of LXRα is strongly increased as compared to healthy controls [56]. This increased expression could be regarded as positive since, in theory, it could lead to an increased efflux of cholesterol from liver cells; however, in the liver, LXR can also induce the expression of genes involved in de novo lipogenesis (see below), thus promoting hyperlipidaemia and steatosis. Moreover, LXR receptors are also involved in the regulatory step of excess cholesterol efflux from peripheral tissues, which could be transported to the liver for its catabolism [44].

#### 1.2.3. Farnesoid X Receptors (FXRs)

Bile acids (BAs) are natural ligands for FXRs that, when activated, translocate into the cell nucleus, form a dimer (a heterodimer with RXR) and bind to responsive elements on DNA, regulating specific target genes involved in lipid metabolism [57]. In particular, FXR is capable of inhibiting the synthesis of BAs by acting on cholesterol 7α-hydroxylase (CYP7A1), the rate-limiting enzyme involved in the bile acid synthesis [58]. A decreased synthesis of BAs from the cholesterol pool causes a liver overload of this molecule within lipid droplets (Figure 3). Therefore, in NASH, the nonfunctional cholesterol efflux from the hepatocytes represents a major cause of its accumulation. A preclinical study by Ma et al. reported that the knockout mouse model for *Fxr* gene (*Fxr*^−/−^ mice) spontaneously developed hepatic steatosis with elevated levels of FFAs and insulin resistance [59], whereas Kunne et al. studied mice with the depletion of hepatic cytochrome P450 reductase (*Hrn* mice), demonstrating that *Hrn* animals developed fatty liver due to the absence of bile acid synthesis [60]. Interestingly, when treated with an FXR agonist, the animals were able to restore their normal bile salt levels, resulting in a significant reduction in TGs within the liver. In humans, a clinical study [61] comparing the expression of genes involved in bile acid metabolism in NASH and NAFL patients found a downregulated FXR protein expression in NASH patients compared to steatotic ones. These findings were also confirmed in pediatric patients [62], where the presence of NAFL or NASH was associated with reduced levels of liver FXR protein.

### 1.3. Gut–Liver Axis in NAFLD

Although a large number of studies has focused on liver metabolism, it has become quite clear that there is indeed a strong bidirectional relationship between the liver and the gut, in particular its microbiota. This relationship, named the gut–liver axis, takes into account not only the microbiota per se but also its metabolites and the possible interactions with genetic, dietary or environmental components, as well as specific hormones released by intestinal mucosa that can have an effect on liver cells, such as FGF15/19, which can interact with hepatic FXF, as shown in Figure 3. Conversely, the liver itself can influence the composition of the microbiota through the secretion of bile acids, IgA and antimicrobial molecules. An alteration in the microbial population, defined as dysbiosis, can lead to the disruption of the intestinal barrier and, in turn, to increased intestinal permeability (“leaky gut”), which allows for bacteria or their products to reach the liver, triggering an inflammatory process. Thus, this sequence of events can play a pivotal role in the development of liver damage, in particular NASH, as nicely demonstrated by Mouries et al. in a NASH mouse model [63]. These alterations in gut permeability have also been detected in NAFLD patients, as assessed by lactulose/mannitol ratio (increased) and claudin-3 and zonulin-1 expression (decreased) [64] or by serum parameters such as diamino oxidase and lipopolysaccharide [65]. Several studies have evaluated the microbiota in NAFLD, with some detecting a reduction in alpha diversity compared to controls [66,67], whereas this difference was not observed in other cohorts [68,69]. Differences at the phylum, genus, or species level have been detected between healthy controls and NAFLD individuals [66,67,68,69,70,71,72], and these alterations correspond to variations in microbiota metabolites [67,70,73,74,75]. The role of these variations in determining liver alterations is supported by several pieces of evidence, such as the ability of microbiota derived from steatotic patients to induce triglyceride accumulation in mice after fecal transplant [70], or the correlation between hepatic gene expression and specific intestinal bacteria [76]. In addition, bacterial products derived from the microbiota can also trigger the immune response in the liver, either through the activation of T [77] or B lymphocytes [78] and thus have a role in determining the severity of the phenotype [72]. There are, however, other variables that can affect microbiota composition, such as the genetic background of the individual or the diet. In fact, the composition of the microbiota can vary according to the presence of polymorphisms in the human genome, as nicely demonstrated by Qin et al. [79]. The presence of SNP in the *PNPLA3* gene has been associated with variations in fecal and plasma metabolomic patterns, in particular with the plasma levels of eicosanoid acid or FA(20:1) [80]. With regard to diet, different effects have been reported in animal models and humans using different diets and/or foods. In patients with NAFLD, starting a restricted diet has modified not only the weight or the lipid content but also the microbiota composition [81,82,83]. Interestingly, the administration of a hypercaloric diet in obese adults induced changes in the microbiota according to the type of diet; in fact, overfeeding saturated fat increased *Proteobacteria*, unsaturated fat butyrate producers, and sugar *Lactococcus* and *Escherichia coli* [84]. Other authors analyzed the different components of the diet, in particular the omega-6/omega-3 ratio in NAFLD subjects; although huge variations in the microbiota were not detected, specific bacteria such as *Catenibacterium* or *Lactobacillus ruminis* were positively and *Clostridium* negatively correlated with n-6 fatty acid dietary intake [85]. Thus, although a huge amount of data has been generated in animal models, further studies in humans are still necessary, with larger cohorts and subcategorization of patients.

### 1.4. Genetic Component of NAFLD

Genome-wide association studies (GWAS) have increased our knowledge of genetic modifications associated with the development and progression of NAFLD, which are important for the assessment of patients’ risk as well as for the possibility of new treatments [86,87,88]. Furthermore, it is useful to detect known or new variants in lean patients or in children, since they represent the minor subtypes of NAFLD [89]. In this regard, the most common gene polymorphisms are related to *PNPLA3* (patatine-like phospholipase domain-containing protein 3 gene), *TM6SF2* (transmembrane 6 superfamily member 2) and *MBOAT7* (membrane-bound O-acyltransferase domain-containing 7). *PNPLA3* encodes the triacylglycerol lipase adiponutrin that mediates the hydrolysis of triacylglycerol both in the liver and in adipose tissue; the mechanisms by which the gene acts in NASH progression are still unclear, as the hepatic overexpression of PNPLA3-I148M in mice showed increased lipid accumulation but no indication of inflammation or fibrosis [90]. BasuRay et al. [86] showed that the variant disrupts the ubiquitination and proteasomal degradation of PNPLA3, leading to accumulation of the mutated protein and impaired mobilisation of TGs from lipid droplets. Moreover, both the PNPLA3 mRNA and protein are upregulated during the activation of hepatic stellate cells (HSCs), so that when the gene is knocked down, the expression of alpha-smooth muscle actin (α-SMA) and collagen type 1 (markers of activated HSCs) is reduced. The variant can therefore increase the profibrogenic effect of the HSCs by promoting chronic damage and fibrosis in the carriers of the mutation [88]. TM6SF2 is a transmembrane protein normally expressed in the liver, kidney, and small intestine that regulates VLDL secretion; this polymorphism is a loss-of-function mutation caused by the substitution of lysine for glutamic acid at position 167 (E167K), which leads to the accumulation of TGs in hepatocytes and, at the same time, lowers systemic lipoprotein levels. Studies in animals with germline deletion of the *TM6SF2* gene have produced conflicting results on VLDL secretion and steatosis, but they have not considered the long-term effects on fibrosis or even the risk of hepatocarcinogenesis [91,92]. Newberry et al. [93] performed a hepatic site-specific mutation of the gene with CRISPR-CAS9 and treated the animals with a profibrogenic (high-fat milk diet) and carcinogenesis-stimulating (DEN) diet and observed how the knockout animals had an accelerated fibrogenesis and carcinogenesis processes compared to the floxed animals. They then administered an AAV8 vector containing either the wild-type gene or the mutant one to *TM6SF2*^−/−^ mice and observed an improvement in carcinogenesis. This is an unexpected result and needs to be further investigated; however, the dosage of the exogenous protein exceeded the normal endogenous levels, so it may have had a beneficial effect regardless of the mutated phenotype.

The *MBOAT7* gene presents a mutation that leads to the replacement of cysteine by threonine as a common variant within the gene. The encoded protein is a lysophosphatidylinositol acyltransferase with specificity for arachidonoyl-CoA as an acyl donor. This protein is involved in the reacylation of phospholipids as part of the phospholipid remodelling pathway. A study published in 2016 [94] found that the rs641738 genotype at the *MBOAT7–TMC4* locus was associated with more severe liver damage and increased risk of fibrosis in patients with NAFLD. In recent years, Thangapandi et al. performed liver-specific deletion of the *MBOAT7* gene and fed animals a chow, high-fat, or choline-deficient diet for 6 weeks; they also genotyped human liver biopsies for the *MBOAT7* mutation and found that either in mice fed a high-fat or methionine-deficient diet and in human samples, the mutation was associated with liver fibrosis, but this did not correlate with inflammation [95]. A meta-analysis by Teo et al. [89], which analysed more than 40 European studies, found that the rs641738C >T variant located downstream of the *MBOAT7* gene was correlated with a more severe hepatic steatosis. Finally, in a clinical trial of NAFLD patients [90], Italian researchers investigated whether there was a correlation between the response to SYBILIN phospholipid-based therapy and the presence of one or more genetic variants and found that the mutated patients did not achieve any improvement compared to those with a normal genotype. In this regard, further studies are needed to understand why these mutations reduce the efficacy of the treatment in NAFLD subjects carrying them. In fact, it is important to genotype patients to ensure proper stratification and to sensitise individuals at higher risk of developing complications, since a recent study observed that the presence of these variants negatively affects the risk of developing cirrhosis and HCC [96]. Unfortunately, there are currently no other clinical trials evaluating the potential impact of NAFLD treatments according to patients’ genetic background.

Although the data are still inconclusive, the genetic component has a role in the modulation of the gut microbiota, as also mentioned above. In a recent preclinical work [97], mice genetically modified carrying the human *PNPLA3* polymorphism (*Pnpla3 ^148M/M^*) and normal mice (*Pnpla3 ^WT^*) were treated with a chow diet or a high-fat diet for 24–52 weeks. After feeding with a high-fat diet, *Pnpla3 ^148M/M^* mice showed reduced microbiota diversity, including an upregulation of the *Firmicutes* species and a downregulation of *Bacteroidetes*, and more fecal bile acids than *Pnpla3 ^WT^* mice. In another work by Lang et al., about 57 patients with NAFLD were genotyped for the *PNLPLA3* mutation and their daily food intake was self-reported for 14 days; after this period, stool samples were sequenced, showing a reduction in two bacterial species, i.e., *Faecalumbacterium* and *Prevotella*, and an increased presence of *Gemmiger* Gram-negative bacteria in *PNPLA3* carriers [98]. It must also be underlined that, in a more extensive study on the gut microbiome, these latter bacteria were associated with a higher risk of hepatocellular carcinoma [99]. In another work by Pirola et al. [100], liver biopsies from patients with NAFLD were analysed to assess the presence of *PNPLA3*, *MBOAT7*, *TM6SF2* gene polymorphisms and a minor variant of *FGF21*. Interestingly, the *Gammaproteobacteria* (including *Enterobacter* and *Pseudoalteromonas*) appeared to be most associated with *PNPLA3* and *MBOAT* mutations, while *Lawsonella*, *Prevotella*, and *Staphylococcus* were most abundant in carriers of the *FGF21* variant. This again shows how genetic mutations influence the composition of the gut microbiota in NAFLD patients.

### 1.5. De Novo Lipogenesis

De novo lipogenesis (DNL) is a process by which circulating carbohydrates are transformed into fatty acids that can be used to form either triglycerides or other lipid species [41]. The transcriptional regulation of the DNL occurs via two main activating pathways, one mediated by the sterol regulatory element 1c-binding protein (SREBP) and the second by the carbohydrate response element-binding protein (ChREBP) [101]. These two pathways are activated by the body’s increased insulin signalling and increased glucose concentrations, respectively. In this regard, it was demonstrated that high carbohydrate and high fructose meals were implicated in the progression from NAFL to NASH [102], as they contribute to the increased energy intake and to insulin resistance. Indeed, increased intracellular carbohydrates levels (mainly glucose and fructose) promote the transition of the ChREBP-α protein from its inactive state to the active one, the passage into the nucleus, and induction of ChREBP-β gene expression [103], promoting the conversion of excess carbohydrates into TGs. In order to study the influence of ChREBP activation on lipogenesis, Hall et al. employed a *Chrebp* gene knockout mouse (*Chrebp*^−/−^), to which they administed a fructose-rich diet. Interestingly, high levels of fructose did not cause hepatic lipid overload but rather a worsening in the inflammatory and fibrotic response. These findings suggest that ChREBP-mediated lipogenesis, in the presence of an excessive intake of fructose, causes a worsening of hepatic steatosis, but at the same time prevents fibroinflammatory status [104]. Therefore, these data indicate the presence of a regulatory balance between DNL induction and liver injury. With regard to clinical evidence, a study by Lambert et al. compared the lipogenesis and the influx of FAs between NAFLD patients and healthy individuals. In this setting, the authors observed that patients with fatty liver had an increase in DNL [105]. The interaction between sugars and lipids is even more complex, since the activity of SREBP is also upregulated by insulin signalling, and its proteolytic activation promotes the transcription of lipogenic genes such as FA synthase and acetyl-CoA carboxylase [106]. In this context, a clinical study showed that SREBP-2 mRNA levels were increased in NAFL/NASH patients compared to controls, and that its expression correlated with the increased levels of accumulated free cholesterol within the liver [107]. In line with these findings, adenoviral-mediated silencing of *Srebp-1* gene in livers of *ob/ob* mice resulted in a significant reduction In the hepatic TG overload, a downregulation of glycolysis, and increased gluconeogenesis [108]. Finally, a hepatic knockout mouse for *Cd36* gene (*Cd36*KO) revealed the attenuation of hepatic steatosis, including a decreased mRNA and protein expression of different lipogenic genes such as *SREBP1c*, suggesting that *Cd36* could have a pivotal role in DNL [41].

NAFLD is usually associated with, and influenced by, other conditions typical of the metabolic syndrome, namely obesity, hypercholesterolemia, and peripheral insulin resistance. At the same time, therapies developed with other primary goals can actually be useful in improving hepatic functionality and lipid status, gaining interest also for the treatment of NAFLD, i.e., incretin agonists (GLP-1 in particular) and THRβ agonists.

## 2. Possible Therapeutic Approaches

### 2.1. Incretins: GLP-1 Agonists

Incretins, namely GIP and GLP-1, are hormones released from the intestine in response to food intake, which mainly stimulate insulin secretion from the pancreas and suppress glucagon. GLP-1 seems to also be active on other tissues and on the intestine itself, since it has anti-inflammatory and trophic properties, which could ameliorate or prevent a condition of leaky gut [109,110,111]. The secretion of incretins usually leads to a reduction of glycaemia and slows down gastric emptying and feedback lost in obese subjects [112]. Thus, GLP-1 agonists have been approved for therapy in T2D (type 2 diabetes) and employed with discrete results to overcome insulin resistance to decrease the appetite and favor weight loss, but they can also be useful in reducing cardiovascular events and mortality, according to a recent meta-analysis [113]. Moreover, secondary effects of weight loss and the improved glycemic control can be beneficial for other tissues like the liver [114]. They were initially studied in combination with metformin, and only lately they were employed alone in NAFLD, since metformin had a limited impact [115]. Different GLP-1 analogues with subcutaneous administration have been developed, namely liraglutide, semaglutide, exenatide, dulaglutide, and cotadutide, with some of them currently in a phase 2 clinical trials for NAFL or NASH. A general view of the influence of these compounds on the liver in NAFL/NASH obtained from clinical trials is reported in Table 1. The efficacy of GLP-1 receptor agonists (GLP-1RA) was recently analysed in a meta-analysis by Wong et al., who reported a reduction in hepatic fat content in patients with T2D and NAFLD treated with GLP-1RA, which also ameliorated hepatic function (ALT, AST, and GGT normalization) and markers of inflammation, also being effective in reducing BMI and visceral adipose tissue [116], results similar to those obtained by Mantovani et al. in their meta-analysis. Interestingly, both groups pointed out that the molecular signalling through which GLP-1 agonists can act on the liver are still elusive [117]. Therefore, studies in vivo have investigated this turning point. Ji et al. suspected decreased oxidative stress in hepatocytes and a lower activation of the hepatic stellate cells due to RAGE and NOX2 signalling in response to liraglutide [118]. Exenatide seems instead to attenuate liver steatosis through the activation of Sirt1 and the downregulation of FABP1, reducing fatty acid uptake in vitro [119]. Liraglutide displayed the potential to ameliorate liver histology in NASH patients from the beginning, improving steatosis and hepatocyte ballooning, with mild gastroenterological side effects [120]. One of the main side effects reported for GLP-1 agonists is nausea, which can favour weight loss, but this, along with the subcutaneous administration, can reduce compliance to therapy and decrease its effects on liver histology [116,117]. Certainly, side effects and the tolerability of the drug need to be taken into account, especially if gastrointestinal symptoms decrease patients’ compliance, as elucidated by Li et al. [121], and in this scenario, the route of administration plays an important role as well. Semaglutide is the only drug that has also been approved for the oral administration route so far. Weekly semaglutide treatment for 24 weeks was even more efficient in improving liver steatosis when combined with firsocostat or cilofexor in a phase 2 trial, even if the combination was less effective in decreasing serum lipids than semaglutide alone [122], whereas daily administration of semaglutide alone for 72 weeks was able to improve NASH in 80% of patients [123] and improve the patients’ health-related quality of life [124]. At the moment, though, it seems that semaglutide is not effective in resolving or ameliorating liver fibrosis once it is present [125], whereas it seems to decrease steatosis and liver inflammation in vivo in *Ldrl^−/−^* mice [126]. Comparative studies and meta-analyses are currently being carried out in order to identify the most effective drug and its posology, which seems to also vary according to the patients’ metabolic status and compliance [127,128,129,130].

Finally, cotadutide is gaining attention recently, as it seems to ameliorate lipid status and liver histology more than liraglutide, probably due to its dual action both as a GLP-1 and glucagon receptor agonist [131], which, according to in vivo studies in DIO mice, is probably due to an impairment in fat deposition through AMPK/mTOR signalling in the liver [132]. With this promising view, other dual agonists have been developed, such as ALT-801, which has just concluded a safety assessment in patients, after improving NAFLD characteristics in vivo more than semaglutide or elafibranor [133], or tirzepatide, a novel dual agonist for GIP and GLP-1, approved for T2D therapy in 2022, which is currently being tested for NAFL/NASH, since it seems to also have beneficial effects on the liver [134]. GLP-1 agonists with double actions, or the combination of drugs with different targets, like liraglutide and elafibranor [135], are therefore an interesting field to explore and exploit for NAFL/NASH treatment.

**Table 1 ijms-24-12748-t001:** GLP-1 analogues studied for NAFLD/NASH so far.

Drug	Mechanism	Clinical Trial Status	Administration Route/Posology	Dosage	Main Liver-Related Outcomes	Clinical Trial Ref	Published Results
Liraglutide	GLP-1 agonist	5 completed; 1 active, not recruiting; 1 recruiting; 1 unknown.	Subcutaneous, 1/day	0.6–1.8 mg/day	↓ Liver volume ↓ Liver fat content ↓ Hepatic de novo lipogenesis ↓ Serum ALT, AST ↓ TG, LDL ↑ Serum HDL	NCT02147925 NCT03068065 NCT01399645 NCT03233178 NCT01237119 NCT05041673 NCT05779644 NCT02654665	[136] [137] [138] [139] [140] [120] [141]
Exenatide	GLP-1 agonist	4 completed; 1 terminated.	Subcutaneous, 2/day or 1/week (extended release)	5–10 μg twice a day	↓ Liver volume ↓↓ Liver fat content ↓ Serum ALT, AST, GGT ↓ Serum LDL	NCT02303730 NCT01006889 NCT01208649 NCT00650546 NCT00529204	[142]
Semaglutide	GLP-1 agonist	5 completed; 2 active, not recruiting; 9 recruiting; 3 not yet recruiting; 1 suspended.	Subcutaneous, 1/week, 1/day OR Oral, 1/day	0.24–2.4 mg weekly; 0.05–0.4 mg daily; 3–14 mg daily (oral tablets).	↓ Liver volume ↓ Liver fat content ↓ Serum ALT, AST, GGT ↓ TG ↓ Fibrosis	NCT03987074 NCT02970942 NCT03357380 NCT03987451 NCT04944992 NCT04216589 NCT05654051 NCT05813249 NCT03884075 NCT03919929 NCT04639414 NCT04822181 NCT05766709 NCT04971785 NCT05016882 NCT05779644 NCT05067621 NCT05195944 NCT05877547 NCT05751720 NCT05424003	[122] [123] [143] [144] [125] [145] [123] [146]
Dulaglutide	GLP-1 agonist	1 completed; 1 not yet recruiting.	Subcutaneous, 1/week	0.75–1.5 mg weekly	↓ Liver fat content ↓ GGT	NCT03590626 NCT03648554	[147]
Cotadutide	Dual GLP-1 and glucacon receptor agonist	2 completed; 1 recruiting; 1 not yet recruiting; 1 terminated.	Subcutaneous, 1/day	300 μg daily	↓ Liver fat content ↓ Serum ALT, AST, GGT ↓↓ Serum cholesterol, TG ↓ Pro-C3 ↓ FIB-4, NFS	NCT03235050 NCT04019561 NCT05364931 NCT05294458 NCT05668936	[131]
Tirzepatide	Dual GIP and GLP-1 receptor agonist	1 active, not recruiting; 1 not yet recruiting.	Subcutaneous, 1/week	0.25–15 mg weekly	↓ Body weight	NCT04166773 NCT05751720	[148] [134]

The words “liraglutide”, “exenatide”, “semaglutide”, “dulaglutide”, “cotadutide”, “tirzepatide”, AND “NAFLD”/“NASH” were searched on www.clinicaltrials.gov, accessed on 19 June 2023. With many of the studies still active, results were not available for each of them at this time. FIB-4: fibrosis-4 index; NFS: NAFLD fibrosis score.

### 2.2. THR-β Agonists

The thyroid hormone (TH) is known to be a key regulator of tissue development and growth, as well as of glucose and lipid metabolism, lowering lipid synthesis and increasing their lysis [149]. In particular, it seems to lower cholesterol levels by increasing receptor-mediated LDL catabolism, increasing ApoA1 levels in HDL and facilitating cholesterol efflux during RCT [150]. Furthermore, T3 can induce cholesterol conversion to bile acids by Cyp7a1 and their excretion trough Abcg5 and Abcg8, in addition to increasing both de novo lipogenesis and fatty acid β-oxidation [149]. For all these reasons, TH supplementation for the treatment of obesity and hypercholesterolemia has been tested, but with adverse effects, principally on bone and cardiac tissues [151,152,153]. Results about TH influence on the liver were obtained in different studies; some of them have correlated low-normal thyroid function with a predisposition to develop NAFLD, since subjects with higher thyroid-stimulating hormone (TSH) and/or fT3 levels, even within “normal” limits, have a higher risk of developing NAFLD [154,155], as well as increased risk of steatosis and liver fibrosis [156] (KIM clinical trial NCT02206841) [157]. Animal models confirmed these observations, also showing that T3 concentration in the liver was lower in case of induced NAFLD [41], and TH deficiency also seems to modify lipid droplet morphology in mouse liver [158]. On the other hand, D’ambrosio et al. did not find a correlation between hypothyroidism and NASH severity in patients [159], whereas Manka et al. correlated a lower level of fT3 with advanced fibrosis in NASH [160]. To overcome the relevant side effects initially seen with TH supplementation, more recent studies are exploiting the different distribution of TH receptors, namely THRα and THRβ. In fact, THRβ is mainly expressed in the liver, which initially led to the development of THRβ agonists to treat hypercholesterolemia [151]. The first evaluated THRβ agonist compounds were CG-1 (sobetirome) and KB-2115 (eprotirome), but clinical trials highlighted an impairment in insulin sensitivity for the first, whereas KD-2115 caused an increase in hepatic enzymes, so the studies were terminated [161,162]. VK2809 (MB07811) is a pro-drug, which is activated when selectively taken up by the liver and cleaved by CYP3A4. In a phase 2 trial, VK2809 was able to decrease more than 30% of hepatic fat content and ameliorate the lipid profile in NAFLD patients after 12 weeks of treatment, as also confirmed 4 weeks after the final dose in 70% of patients (NCT02927184) [163]. Currently, a phase 2b trial called VOYAGE is ongoing and will evaluate VK2809 efficacy after 52 weeks, also evaluating the efficacy of different posology on fibrosis (NCT04173065). MGL-3196 (resmetirom) is the other promising THRβ agonist. Resmetirom has just concluded a phase 2 double-blind trial in which the goal of a reduction of more than 30% of hepatic fat fraction was reached in 68% of treated patients after 36 weeks, as determined using MRI [164]. A general improvement was also observed in the lipid profile (LDL, ApoB, VLDL), in markers of liver injury (ALT, AST, GGT) and fibrosis (PRO-C3), without affecting the bodyweight, whereas side effects were not relevant. These clinical observations were confirmed in a mouse model of Diet-induced NASH by Kannt et al., in which resmetirom had similar effects both in liver fat content and in lipid profile [165]. Wang et al. recently determined that the action of resmetirom is also mediated by the suppression of STAT3 and NFkB signalling through the activation of RGS5 in mice, suggesting a new collateral target for NASH treatment [166]. A promising phase 3 clinical trial with resmetirom is currently ongoing (MAESTRO-NASH NCT03900429), and the treatment efficacy on fibrosis will also be evaluated after 52 weeks [167]. Furthermore, to minimize side effects related to the remaining affinity with THRα, the MGL-3196 compound has recently been optimized, and it seems that its newer form, compound 15, has higher potency and affinity for THRβ, with a liver-to-serum ratio of 93:1 compared to the 6:1 ratio of MGL-3196 [168]. Karim et al. nicely reviewed the ongoing studies on resmetirom in NAFLD including ASC-41 and TERN-501, two compounds recently developed that can be orally administered, and are currently in phase 2 studies (NCT05462353, NCT05415722) after having shown promising results in rodents [169,170].

Other molecules activating THRβ signalling have been tested with discrete results in animal models, such as TG68, which ameliorates hepatic steatosis and lipid metabolism, mainly improving mitochondrial fatty acid oxidation downstream the activation of THRβ [171], or KBP-089, a double-receptor agonist for amylin and calcitonin, a hormone secreted by thyroid C cells, which seems to ameliorate the lipid profile, especially when in combination therapy with liraglutide [172,173], but these studies are still at their beginnings. 

### 2.3. PPAR Agonists

#### 2.3.1. PPARα

PPARα is expressed ubiquitously but is abundant in the liver; in the hepatocytes, it triggers fatty acid oxidative metabolic pathways in peroxisomes, mitochondria, and microsomes, increases lipoprotein metabolism, whereas it decreases lipid deposition and inflammation. In Kupffer cells and in hepatic stellate cells, its activation reduces inflammatory response. In all the three different liver cell types, PPARβ/δ activation reduces inflammation, but in hepatocytes it can also reduce lipogenesis. PPARγ, although highly expressed in adipocytes, is also present in hepatocytes and stellate cells, where it plays different roles. In hepatocytes, it triggers lipid deposition, but it also reduces the production of inflammatory mediators, whereas it reduces collagen production, cell proliferation and migration in stellate cells. Due to the possible positive effects on disorders characterized by lipid deposition and inflammation such as NAFL or NASH, several molecules, either agonist or antagonist of the various PPAR subtypes, have been developed and initially tested in in vitro or in vivo models. PPARα increases the uptake of FAs through the upregulation of FATP and CD36, as well as the transformation to acyl-CoA via an increased expression of acetyl-CoA synthetase. After FA uptake, PPARα regulates the transport of FAs into mitochondria and promotes their β-oxidation. This in turn leads to a reduction in the synthesis of FA and TG, and then to a decrease in very low density lipoprotein (VLDL). PPARα can also induce the expression of ApoA-I and ApoA-II mRNA, thus increasing HDL in the circulation, and reducing the atherosclerotic risk. The ability to reduce inflammation, as demonstrated in macrophages, is due to the capacity of PPARα to inhibit the migration of nuclear factor kappa B (NF-κB) subunit p65, as well as to reduce the phosphorylation level of c-jun subunit AP-1. This, in turn, reduces the production of proinflammatory cytokines, including IL-1 and TNF-α [174].

Among activators of PPARα, we should quote fibrates and elafibranor. Fenofibrate has been employed in various animal models of NAFL/NASH, demonstrating its ability to improve lipid metabolism, reduce hepatocyte damage (usually assessed by AST and ALT dosage), oxidative stress, and also collagen deposition [175,176,177]. Initial trials with fenofibrate showed a decrease in serum lipids and transaminases [178,179,180], but no change in steatosis, lobular inflammation, or fibrosis [178]. Moreover, a double-blind clinical trial performed on subjects with NAFLD demonstrated a reduction in serum triglyceride levels but also detected an increase in liver fat content [181]. Various researchers have tried to modify the chemical characteristics of fenofibrate, in particular using nanoparticles [182,183], obtaining good results in mouse models of NAFLD, with an important reduction in liver lipid concentration and oxidative stress [182,183]. Another activator of PPARα is pemafibrate, which has been recently evaluated in a large double-blind, randomized trial for its ability to reduce cardiovascular risk; although it was able to reduce plasmatic triglycerides and VLDL cholesterol, there was no difference in the number of cardiovascular events in the treated group that, however, showed a higher frequency of adverse renal events. The authors also mentioned that, in the treated group, there was a reduction in the frequency of NAFLD [184]. Two other studies reported an improvement in serum parameters, i.e., lipids and transaminases, in subjects treated with pemafibrate, but these studies were retrospective and with a limited number of subjects [185,186].

A different compound, a PPARα/δ agonist, was also developed, and named elafibranor (initially called GFT505). The clinical trials evaluating the efficacy of this compound in NASH patients produced contradictory results; an initial trial that lasted a year showed a decrease in liver enzymes, lipids, and glucose levels and an increase rate of NASH resolution without fibrosis worsening in treated patients, although this late result was barely significant [187]. A phase 3 study, however, failed to show any difference in the treated group in the ad interim analysis, a fact that led to the interruption of the study.

Another PPARα/γ agonist, saroglitazar, provided encouraging results in vitro and then in animal models, with a reduction in lipid deposition as well as collagen and α-SMA production in a NASH mouse model [188]. These data were further confirmed in a different NASH model (rats fed a high-fat diet), in which the authors detected a reduction in inflammatory mediators such as IL-6, TNF-α, TGF-β, plus a reduction in liver lipid content [189]. In humans, a phase 2 randomized, double-blind trial evaluating the effect of saroglitazar on NAFLD patients showed a reduction in liver enzymes at 16 weeks, but also a significant decrease in liver lipid content and an initial reduction in liver stiffness [190]. These data were confirmed in other nonrandomized studies performed in India, in which a reduction in plasma lipid parameters and liver stiffness was observed [191,192,193]. It must be noted, however, that in one of these studies [192], the reduction in liver stiffness parameters was present only in the treated patients that showed a reduction in body weight >5%. Further data will be necessary, and clinical trials are currently undergoing (see Table 2).

#### 2.3.2. PPARγ

PPARγ is an important differentiation factor, which is mainly expressed in adipose tissue; however, it seems to also play an important role in NAFL/NASH, since it can act both on lipid metabolism and on the inflammatory response. The first effect is due to PPARγ’s ability to increase the outflow of cholesterol via ABCA1/LXR pathway, whereas it can curb inflammation binding a specific element present in the promoter of *NF-κB*, *STAT1*, and *AP-1* and block their transcription. In activated macrophages, increased PPARγ expression can reduce the secretion of cytokines such as IL-1β and TNF-α and this, in turn, can prevent the switch from quiescent to activated hepatic stellate cells [49,194]. The PPARγ agonist that has been evaluated quite extensively is pioglitazone, a drug normally employed in the treatment of type 2 diabetes. An initial study detected a reduction in liver enzymes in NASH patients, with an improvement in histological parameters and a reduction in liver lipids and plasma triglycerides [195,196], and subsequent trials confirmed these data in NAFLD and type 2 diabetes patients [197,198] and in NASH subjects [199]. These data thus seem encouraging, but the presence of side effects of the drug must be underlined, namely an increase in body weight and the presence of edema. To overcome this problem, an international group recently reported a modified pioglitazone molecule with the same efficacy but with less side effects if compared to the original drug [200,201]. This molecule is a deuterium-stabilized version of pioglitazone; the deuterium modification increases the percentage of the R isomer, actually reducing its PPARγ activity but maintaining its effects of other enzymes, such as the mitochondrial pyruvate carrier. Interestingly, both in an animal model as well as in a phase 2 trial, this molecule was able to improve liver histology without weight gain or edema, side effects that have been attributed to a pure PPARγ action (Table 2).

As described above, each PPAR could play a different role in the prevention/resolution of NAFL/NASH, either acting on lipid deposition or inflammation and collagen deposition. Thus, in theory, a molecule able to trigger the activation of all the three PPARs could have a better effect on liver histology. Recently, a pan-PPAR agonist was developed, lanifibranor. Its use in vitro and in vivo on different models of NASH (choline-deficient + high-fat diet or Western diet) showed that lanifibranor was able to reduce liver lipid content, fibrosis, and macrophage infiltration in both models [202]. Interestingly, the effect of lanifibranor was superior to the effect of specific PPARα, δ, and γ agonists (namely fenofibrate, GW01516, and pioglitazone) in the same models, thus supporting the idea that the action on multiple PPARs could have a synergic effect. Lanifibranor has also gone through phase 2 trials [203,204], evaluating its efficacy in NASH patients. The randomized double-blind trial showed a significantly higher percentage of patients with inflammation or fibrosis improvement in the group treated with the highest dose of lanifibranor; however, side effects (weight gain, edema, anemia) were also reported in the treatment group (Table 2).

### 2.4. FXR Agonists

FXR, similarly to PPARs, forms a heterodimeric complex with RXR⍺; the heterodimer binds to AGGTCA-like motifs, regulating the expression of genes involved in bile acid metabolism, but also in lipid synthesis and transport. FXR is activated by bile acids, and it is present in both the liver and the intestine, being involved in enterohepatic circulation. In the liver, FXR induces the expression of several genes involved in the uptake, conjugation, and secretion of bile acids, but also of MDR2/3, which excretes phospholipids to facilitate bile acid secretion into the bile [205]. FXR activation can affect hepatocyte lipid balance, promoting reverse cholesterol transport through the induction of SR-B1, but also inhibiting hepatic CD36 and lipogenesis enzymes [206]. In addition, it can also inhibit the expression of apolipoprotein A-I and reduce HDL production [207], as well as decrease the expression of apolipoprotein CIII [208]. It must be underlined, however, that there are four FXR isoforms, α1 to α4, which differ in their activation function domain 1 at the N-terminus and by the presence of four extra amino acids (MYTG) of the DNA binding domain. The four isoforms are generated by alternative promoter usage and alternative splicing, and the two promoters are differentially active among tissues; human liver mainly express FXRα1 and α2 isoforms [209,210], and the ratio between these two isoforms is important for lipid metabolism. In fact, FXRα2 can promote hepatic lipid clearance, since it binds to a specific DNA motif that regulates various metabolic effects, including a reduction in de novo lipogenesis. This is a pivotal process in NAFL/NASH pathogenesis, since its contribution to lipid accumulation has been reported increased up to six times in NAFLD [105].

Due to the FXR activation effect on lipid metabolism, FXR agonists could represent a therapeutic option not only for disorders such as primary biliary cholangitis, but also for NAFL/NASH. Among FXR agonists, some have already entered the clinical trial phase or clinical practice (obeticolic acid, ciloflexor, tropifexor), whereas others have only been tested in animal models. Obeticholic acid is a potent selective FXR agonist, currently approved for PBC patients as a second-line treatment [211]. Due to its possible therapeutic application in NAFL/NASH, several studies have been performed to evaluate obeticolic acid efficacy in different animal models [212,213,214,215,216,217,218], showing a reduction in liver lipid content, oxidative stress, and fibrosis. The results obtained from obeticolic acid in clinical trials on NASH noncirrhotic patients showed a significant improvement in NASH score, fibrosis, lobular inflammation, and steatosis in the treated group [219]. However, the treatment had several unwanted side effects, including pruritus (in about 30% of the case) and significantly increased serum LDL cholesterol and decreased HDL cholesterol. A phase 3 clinical trial, testing two different dosages (10 or 25 mg/day), showed significant improvement in fibrosis [220,221], but in this case, the appearance of pruritus in the higher dosage group affected about 50% of the patients. These results were not sufficient to grant accelerated FDA approval, since the committee declared that benefits of obeticolic acid do not outweigh the potential risks, in particular regarding drug-induced liver injury. Cilofexor is a selective nonsteroidal FXR agonist that primarily activates FXR in the intestine and does not undergo enterohepatic circulation. Activation of intestinal FXR causes the release of fibroblast growth factor 19, that, through the portal system, reaches the liver and binds to the FGFR4/β-Klotho receptor complex, initiating a cascade that inhibits bile acid synthesis from cholesterol [222]. In addition, FGF19 reduces the expression in the hepatocytes of sterol regulatory element-binding protein 1c, the master regulator of de novo lipogenesis [223]. After positive results in an animal model [224] and in phase 1 [225], ciloflexor was tested in noncirrhotic NASH patients with two different dosages in a phase 2 clinical trial, i.e., 30 and 100 mg/day, showing a significant reduction in liver lipid content assessed using MRI and in liver stiffness only in the higher dosage group [226]. Even in this case, however, about 20% of the patients experienced moderate to severe pruritus. For both these two FXR agonists, pruritus represents an important side effect, and this could be related to increased production of IL-31, as recently reported by by Xu et al. [227], who detected a significant dose-dependent increase in IL-31 levels in NASH patients treated with ciloflexor, probably due to a direct effect of the FXR. Tropifexor is a nonbile acid FXR agonist [228] that reduced fibrosis inflammatory markers and lipid liver deposition in two different NASH animal models [229]. The drug developer also tested it in subject with various degrees of hepatic impairment [230], and recently, the results of phase 2a/b trials were published [231]. In this paper, the authors reported the results obtained in different groups of NASH patients treated with various doses of tropifexor, showing a reduction in liver enzymes and lipid content, in particular with the highest dosage, but also, this drug caused pruritus in treated individuals. A histological evaluation was performed only in the patients treated with the highest dosage of the drug, and significant differences were detected when employing fluorescence microscopy and artificial intelligence for the analysis of the slides, showing a reduction in fibrosis in the perisinusoidal areas in particular [232].

The presence of side effects in the drugs described up to now (see Table 3) has prompted several researchers and companies to develop new molecules belonging to different drug categories. Several of them are new FXR agonists, and they have mainly been tested in vitro or in animal models. Here, we decided to report only molecules that have already been employed in clinical trials or those that present new particular characteristics. Among the new FXR agonists, three of them (namely EDP-305, MET409, and vonafexor) have already been employed in clinical trials with favorable results, mainly a reduction in liver fat content [233,234,235]. For all of them, however, pruritus was reported as a side effect, being more prevalent with increasing doses of the drug. Other new FXR agonists that have been developed present a different characteristic, since they do not act only on FXR but also on a second molecule involved in the pathogenesis of NASH. These include a dual activator of FXR and G protein-coupled receptor GPB AR1, which showed good results in a NASH mouse model with a strong reduction in steatosis and fibrosis [236,237], results similar to those obtained by an activator of FXR and transmembrane G protein-coupled receptor 5 (TGR5) [238,239]. A different approach was taken by other researchers; as previously described, FGF-19 could be released from the intestine and act on the liver to decrease gluconeogenesis. Harrison et al. employed aldafermin, an FGF19 analog, to treat NASH patients [240,241], detecting a significant reduction in lipid liver content, but also a higher number of treated patients that showed fibrosis reduction and NASH resolution. Another way of tackling the problem would be the use of a combination of drugs influencing different metabolic pathways, and several combinations have been employed up to now; in some cases, the use of two different molecules reached a better result [242], or reduced the side effects [243], whereas in some others, no significant improvement compared to single therapy was observed [244].

### 2.5. Natural Compounds

The possibility of combining different compounds should also consider the use of “natural” molecules that could improve liver lipid metabolism, and apolipoproteins can be considered among the potential therapeutic targets for NAFL/NASH. Several works focused on VLDL and LDL lipid profile of NAFLD patients [245,246]; however, the role of HDL remained secondary and poorly characterised. Serum analyses of individuals affected by NAFLD showed reduced levels of polyunsaturated phospholipids and FFAs, which were negatively correlated with BMI, IR, TGs, and hepatocyte ballooning [247]. It is clear that HDL composition can also influence lipid metabolism and liver damage. Indeed, the loss of the atheroprotective role of HDL leads to reduced RCT efficiency, a fact that was already reported in a study in which NAFLD patients had a dysregulation of cholesterol efflux compared to healthy controls [248]. Following this line, one of the proteins that have been implicated in NASH pathogenesis is apolipoprotein A-I (APOA-I, encoded by the *APOA1* gene, HGCN:600). APOA-I is the main protein component of high-density lipoproteins (HDL), which are in charge of the transport of cholesterol from the peripheral tissues to the liver for catabolism or recycling [249]. It has been recently shown that APOA-I levels are reduced in patients with NAFLD [195,250,251], and lower APOA-I levels were associated to NAFLD risk in a Korean population [252]. Thus, in theory, increased expression of APOA-I could have a positive effect on the liver in presence of NAFL/NASH. Indeed, in vitro experiments on hepatocytes showed that overexpression of APOA-I was reported to reduce hepatic endoplasmic reticulum stress and lipogenesis in hepatocytes [253], but a reduction in lipid content was also observed in a NASH mouse model treated with adenoviral vector containing human APOA-I [254,255]. Interestingly, there is a naturally occurring variant of APOA-I, namely APOA-I_Milano_, which has stronger antiatherogenic capacity [256]; its use in an APOA-I knockout mouse was able to completely revert the NASH phenotype [257]. Although these data provide the proof of concept that APOA-I could be extremely helpful in NASH treatment, its use in animal studies or clinical trials has been hampered by the difficulty in producing the recombinant form of this apolipoprotein. We recently reported the use of genetically engineered edible plants to simultaneously obtain an effective system of production and delivery of recombinant APOA-I_Milano_, which was orally administered to high-fat diet-fed B6 *Apoe^−/−^* mice, a strain of mice that progressively develop atherosclerosis and NASH after 7 weeks of high-fat diet exposure [258]. Although further data are required, the treatment caused a reduction in CD68-positive cells in the liver, suggesting a hepatic anti-inflammatory that could possibly lead to NASH attenuation [259].

There are hundreds of reports about the experimental testing of therapeutic potential for NAFL/NASH of natural compounds, based on, or related to, lipid metabolism targeting, but for the purpose of this review, we focused on those that reached the preclinical stage of development (at least in rodents) and on those related to lipid transport and metabolism.

Polyphenols, plant-derived molecules, have been extensively investigated in different contexts for their antioxidant and anti-inflammatory properties. Ferulic acid and P-coumaric acid were evaluated for hepatoprotective effects in combination, and their synergistic action has been reported to decrease lipid liver content in a high-fat diet (HFD)-fed mouse model of NAFLD, finding an association with decreased expression of transporters involved in free fatty acid uptake [260]. Another plant-derived molecule, andrographolide, a diterpene lactone found in a traditional Chinese herbal medicine, was reported to be effective in reducing obesity and steatosis in an HFD-fed mouse model and, interestingly, it was reported to have a similar action of reducing free fatty acid uptake in hepatocytes [261]. Several other plant-derived molecules were tested for hepatoprotection in NAFL/NASH mouse models and were found to exert such protection by reducing lipid accumulation in the liver. Yangonin, a bioactive kavalactone extracted from a tropical plant (kava), has been shown to attenuate NAFLD induced by a high-fat diet in mouse models with specific regard to reduction in lipid accumulation [262]. This action was reported to be the result of the modulation of lipogenesis via the activation of FXR and its downstream genes, since siRNAs directed against FXR were able to abrogate the effect. Another plant-derived molecule, trans-chalcone, was reported to reduce lipid accumulation in an NAFLD model of high-fat emulsion-fed rats. Trans-chalcone treatment, via the modulation of relevant microRNAs (miR-34a, miR-451 and miR-33a), increased hepatic cholesterol efflux through ABCA1 upregulation and reduced liver inflammation [263]. 3-Acetyl-oleanolic acid (3AcOA), a molecule derived from plant-derived oleanolic acid, was reported to ameliorate hepatic parameters in an HFD rat model and to reduce lipid accumulation in primary hepatocytes and HepG2 cells [264]. Niga-ichigoside F1, a plant metabolite extracted from *Geum japonicum* Thunb. var. *chinense*, was found to reduce hepatic steatosis and lipid accumulation in an HFD-fed mouse model. The reduction in lipid accumulation was associated with NRF2 nuclear translocation, and even in this case, silencing of NRF2 abolished the effect [265]. Another plant-derived molecule, calycosin, a 7-hydroxyisoflavone, was tested in a methionine choline-deficient diet NASH mouse model, in which it was able to induce a reduction in hepatic fibrosis and in liver triglyceride content. It is interesting to note that, even in this case, the FXR pathway was involved, since various FXR downstream target genes involved in triglyceride synthesis were downregulated, whereas increased PPARα and β-oxydation activity was detected [266], and similar results were obtained in an NAFLD mouse model [267]. Alisol B, a triterpenoid, was also found to reduce hepatic steatosis and fibrosis in mice in which NASH was induced by a high-fat diet plus carbon tetrachloride (DIO + CCl4) and choline-deficient and amino acid-defined (CDA) diet. Alisol B also reduced lipotoxicity and lipid accumulation by suppressing CD36 expression [268]. Neomangiferin, a bioactive constitutive of *Mangifera indica* leaves and *Rhizoma anemarrhenae*, has been shown to reduce liver lipid accumulation and triglycerides in an HFD-fed rat model [269]. When molecules with potential therapeutic effects had low bioavailability, researchers combined biotechnological approaches to increase such bioavailability via the oral route. This is the case of sylimarin, a potent hepatoprotective agent, that was loaded on lipid–polymer hybrid nanoparticles containing chitosan (CS-LPNs) to increase oral bioavailability. The CS-LPNs were tested in an NAFLD hyperlipidaemic rat model, resulting in a significant reduction in lipid accumulation in the liver [270].

#### 2.5.1. Natural Compounds in Trials for NAFL/NASH

As already mentioned, there are many natural bioactive substances that have been tested in a preclinical setting, both in vitro and in vivo, which can have an effect alone or in combination with further treatments. However, the compounds that have yielded the most significant results are tocopherols, silymarin, caffeine, resveratrol, and curcumin. One of the substances recommended for the combination treatment of NAFLD is vitamin E (alpha-tocopherol), which is able to keep blood lipid levels under control because it contains fibres that reduce the absorption of cholesterol in the intestine. The results of these studies show an important contribution of the molecule to processes that prevent steatosis and liver damage, as well as to biochemical parameters commonly used in the clinics (HDL/LDL/AST/ALT/γGT).

##### Vitamin E

In 2006, a clinical trial in children with NAFLD treated with vitamin E and vitamin C and subjected to lifestyle changes found no difference in the reduction in ALT and insulin resistance levels. However, the interindividual variability was very high and, more importantly, physical activity and diet alone led to an improvement in liver function independent of antioxidant assumption [271]. Other multicentre clinical trials evaluated the efficacy of vitamin E both alone [272] and in combination with spironolactone [273] in patients with simple steatosis or NASH. In the latter case [273], the level of Noggin protein was lower in simple steatosis in comparison with controls; in the former case, however, the presence or absence of certain polymorphisms for response to vitamin E was assessed in all subjects using q-PCR, showing that the Chinese population responds better to treatment due to the presence of such polymorphisms. A more recent study [274] evaluated the minimum effective dose of vitamin E required to reduce steatosis in NAFLD patients, showing that there was no association between the observed clinical effect and the treatment dose. However, we have to consider that in NAFLD patients, vitamin E is sequestered in lipid droplets, preventing proper antioxidant delivery; therefore, it is more difficult to assess the dosage referred to the hepatic overload [274].

##### Silymarin

The efficacy of silymarin was evaluated in 2017 in a double-blind clinical trial in patients with biopsy-proven NASH over a one-year period, and a high percentage of treated patients showed reduced fibrosis, as assessed by histological analysis and a significantly lower AST/platelet ratio [275]. More recently, a multicentre phase II clinical trial of silymarin at two different doses showed a nonsignificant improvement in liver steatosis and no differences in biochemical parameters (ALT, AST) and NAS. In addition, gastrointestinal adverse events were reported in 56% of the participants, although serious events occurred in only 5% of the cases [276]. The protocol of the SILIVER trial [277], which is still in the recruitment phase, has also been published. In this case, patients are treated with silymarin and vitamin E capsules, and any change in the degree of liver steatosis is assessed using CT or MRI, as well as serum levels of γGT, ALP, and glycated hemoglobin.

##### Resveratrol

With regard to resveratrol, in 2014, a clinical trial on NAFLD patients, who underwent diet and physical activity, showed that the treatment with the natural compound had a significant effect on AST levels, anti-inflammatory cytokines, and steatosis after 12 weeks, demonstrating that the treatment overcame the lifestyle changes adopted by the placebo group [278]. On the other hand, another placebo-controlled clinical trial [279], in which higher doses of resveratrol were tested, showed that the treatment had no effect on hepatic lipid content or VLDL and TG secretion and hypothesised that, probably, this is due to the fact that only 16 male patients were recruited in this study and that dosage is still an important issue for the design of a clinical trial [279].

##### Curcumin

One natural substance that has become very fashionable in recent years and is of interest in both clinical and preclinical studies is curcumin. The trials listed on clinicaltrials.gov report the use of capsules or tablets of curcumin alone or of phospholipids and curcumin together (see Table 4). In particular, trials evaluating the efficacy and safety of curcumin in both NAFLD and obese patients at different doses [280,281] assessed both the reduction in fibrosis by the use of Fibroscan [280] and serum levels of inflammatory markers or fat/glucose parameters [281], but these results were not significantly different from those observed in the placebo group.

##### Caffeine

A few years ago, a pilot clinical trial recruited patients with T2DM and treated them with caffeine and chlorogenic acid in combination or alone, thus excluding some coffee components. Not surprisingly, patients who received the combination of the two compounds showed significant weight loss compared to all other groups, with an interesting increase in *Bifidobacterium* species per gram of analysed feces [282]. More recently, the same group reported the results of a trial (not registered on clinicaltrials.gov) involving two centers in which patients with T2DM and NAFLD were supplemented with a daily dose of caffeine, chlorogenic acid alone or in combination. The only reported difference with the placebo group was the reduction in total cholesterol for the caffeine group and the increase in insulin levels in the caffeine plus chlorogenic acid group, with no effects of the treatment groups in reducing hepatic outcome [283].

**Table 4 ijms-24-12748-t004:** Selected natural compounds in trial for NAFLD/NASH.

Compound	Identifier Trial	Condition	Drug	Dosage	Status	Citations
Vitamin E	NCT02690792	NAFLD	Vitamin E	200 IU capsule, two capsules each morning and each evening every 4 months	Completed	-
	NCT01792115	NAFLD	Vitamin E	200–400–800 IU/day for 24 weeks + optional extension for up to 120 weeks	Completed	[274]
	NCT02962297	NASH	Vitamin E	Softgel 100 mg, oral adiministration for 96 weeks.	Completed	[272]
	NCT04198805 (PUVENAFLD)	NAFLD	Vitamin E and DHA-EE	Vitamin E 1000 mg capsules/DHA EE (1.89 g) alone or in combination once daily for 6 months	Completed	Results posted
	NCT01147523	NAFLD	Vitamin E and spironolactone	Spironolactone, tablets, 25 mg and vitamin E capsules 400 mg daily, for 52 weeks	Completed	[273]
	NCT01934777	NAFLD (4–16 years)	Docosahexaenoic acid plus vitamin E plus choline	DHA 250 mg plus vitamin E (39 UI) plus choline 201 mg per day for 6 months	Completed	-
	NCT00655018 (VITENAFLD)	NAFLD (3–20 years)	Vitamin E, vitamin C	a-tocopherol 600 IU/d plus ascorbic acid 500 mg/d	Completed	[273]
Resveratrol	NCT01464801 (LIRMOI3)	NAFLD	Resveratrol	Resveratrol 500 mg tablet, three times daily for 6 months	Completed	-
	NCT02030977	NASH	Resveratrol	Resveratrol 500 mg/d, one capsule per day for 12 weeks	Completed	[4]
	NCT01446276	NAFLD	Resveratrol	500 mg, three times daily for 6 months	Completed	[279]
	NCT02216552	NAFLD/IR	ResVida (13–18 years)	Resveratrol 75 mg twice daily for 30 days	Completed	-
Silibilin	NCT05497765	NAFLD	Silibilin	Silibilin extract, four tablets with warm water twice a day	Recruiting	-
	NCT04640324	NAFLD	Silybin–phospholipids, vitamin D, vitamin E	303 mg of silybin-phospholipid complex, 10 ug of vitamin D, and 15 mg of vitamin E, twice a day for 6 months	Completed	[284]
Ginger	NCT02289235 (GinLivDM)	NAFLD/ T2DM	Ginger	Ginger capsule 1000 mg twice daily for 3 months	Completed	-
Berberine	NCT04049396	NAFLD	Berberine	6.25 g/day berberine for 6 weeks	Completed	-
	NCT05523024	NAFLD	Berberine	1500 mg of berberine extract per day in three doses.	Recruiting	-
	NCT03198572	NASH	Berberine	Tablets 0.5 g, 30 min before each meal, for 48 weeks	Unknown	-
Silymarin	NCT03749070 (The SILIVER Trial)	NAFLD	Silymarin	700 mg of silymarin, 8 mg Vitamin E and 50 mg phosphatidilcholine, daily for 12 weeks	Recruiting	[277]
	NCT02006498	NASH	Silymarin	700 mg, administered three times daily for 48 weeks	Completed	[275]
	NCT00680407	NASH/HCV	Legalon	420 or 700 mg dose of silymarin (five pills, three times daily) for 48–50 weeks	Completed	[276]
	NCT05051527	NAFLD	Legalon	140 mg of silymarin for 6 months	Recruiting	-
Caffeine	NCT02929901	NAFDL/ T2DM	Caffeine	caffeine (200 mg)/chlorogenic acid (200 mg) or combination of the two, one capsule/day for 6 months	Completed	[282]
Curcumin	NCT02908152	T2DM/NAFLD	Curcumin capsules 1.5 g/d	1.5 g/die	Completed	[280]
	NCT03864783	NAFLD/IR	Phospholipid curcumin capsules	One tablet (200 mg curcumin) twice daily for 3–4 days	Completed	[281]
	NCT04315350	NAFLD/IR	Curcumin tablets	One tablet (100 mg curcumin) twice daily + prednisolon 50 mg every morning	Recruiting	-
	NCT02369536	NAFLD	Fish oil 70%, DHA, silymarine, curcumin, D-a-tocopherol, choline bitartrate, phosphatidilcoline in sunflower oil	Two soft gelatin capsules of 800 mg per day	Completed	Results posted
Mastiha	NCT03135873 (MAST4HEALTH)	NAFLD	Mastiha	2.1 g/day for 6 months	Completed	[285]

The words “Non-alcoholic fatty liver disease” and “Tocopherol ”, “resveratrol”, “curcumin”, “sylmarine”, “berberine” and “caffeine” were searched on www.clinicaltrials.gov, accessed on 17 July 2023.

#### 2.5.2. Challenges in the Use of Natural Compounds

Natural substances in trial for NAFLD are often combined with lifestyle changes (diet and physical exercise) and taken in combination with drugs or other natural products. Therefore, several aspects need to be considered, including the route of administration, the dosage and the timing of the treatment, similarly to synthetic drugs used in clinics. As shown in Table 4, there are several completed clinical trials whose results are not published, so it is often difficult to draw concrete conclusions about the efficacy and toxicity of treatment. In addition, each study focuses on a different clinical condition, which can range from simple steatosis to a more inflammatory and fibrotic state (NASH), or include T2DM patients, who in 70% of cases also have liver steatosis. Usually, expected outcomes are defined for each study according to the clinical status of the patients, so that the evaluations are closely related to the efficacy of that specific compound, at a specific dosage, with a particular schedule of administration, associated with the selected patient’s cohort. The latter is a crucial and at the same time very limiting aspect, as it is restricted to what is expected to be observed. Other issues to consider in the design of clinical trials are the interindividual variability and the bioavailability of natural compounds, which leads to a lack of specific organ targeting. In addition, the recruitment of an adequate number of patients is essential to obtain reliable results. Finally, most of the adverse effects of metabolic syndrome are associated with complications related to obesity, T2DM, and cardiovascular disease, but lipid overload can also affect the liver and NAFLD, cirrhosis, and, ultimately, hepatocellular carcinoma can be consequent to many processes related to metabolic syndrome (MALFD). Obviously, the use of these natural substances can be quite different from conventional treatments, as the use of natural components may seem less restrictive than the use of a drug with major side effects, but this is not always the case. In fact, following the example of curcumin, gastrointestinal side effects such as constipation and abdominal pain have been reported in some subjects in NAFDL and T2D clinical trials, and although they are not as serious, they represent an obstacle to the well-being of the patient [280].

### 2.6. Obstacles and Challenges in Clinical Trials

Although we have reported a large number of clinical trials, either with published results or still ungoing, bigger and better cohorts are still needed, mainly due to the large differences that can be present in patients classified under the general definition of NAFLD. In fact, a more precise subcategorization could allow for obtaining a better evaluation of the results, since a particular drug or natural compound could possibly have an effect in the initial phase of the disease and not in later ones. It must be underlined that in NAFL, lipid deposition is predominant, whereas in NASH, the inflammatory process prevails, leading to fibrosis deposition. In addition, even the level of fibrosis can be quite different among patients, and this grading should be taken into account when assessing the efficacy of the therapy in particular. Moreover, the presence of specific polymorphisms in genes involved in lipid matabolism could also affect the response to the administered drug, and thus genotyping the enrolled subjects could provide better subcategorization of patients, in particular assessing whether a specific subgroup could be a nonresponder. These considerations should thus be added to those reported above for natural compounds, also considering that a combined therapy (different drugs or drugs and natural compounds) will probably be the best way to treat these patients.

## 3. Materials and Methods

The following words, accompanied by “NAFLD” and/or “NASH” were searched in PubMed, and articles considered relevant for the topic were included. Lipoproteins; reverse cholesterol transport; peroxisome proliferator-activated receptor; PPAR; farnesoid X receptor; FXR; GLP-1 receptor agonist; incretins; liraglutide; semaglutide; exenatide; dulaglutide; tirzepatide; cotadutide; THRβ; thyroid hormone; GC-1; VK2809; resmetirom; MGL-3196; TG68; lanifibranor; saroglitazar; cilofexor; tropifexor; obeticholic acid; EDP-305; natural compounds; APOA-I; polyphenols; tocopherol; resveratrol; curcumin; silymarin; berberine; and caffeine. Ongoing clinical trials were looked up in www.clinicaltrial.gov and the specific date of accession is reported in the text.

## 4. Conclusions

NAFLD and NASH constitute a health problem with a great impact on society, also considering the possible progression to hepatocellular carcinoma. For this reason, it is mandatory to identify possible therapies that, associated with weight loss, could lead to an improvement in liver histology, in particular reducing steatosis and fibrosis. Among the possible therapies, those tackling lipid metabolism surely deserve attention, since they can modify or revert the processes involved in lipid accumulation. In addition, due to the strong interaction between NAFLD and metabolic syndrome, it should also be kept in mind that, in some patients, the therapeutic approach could also involve a reduction in insulin resistance. Drugs and natural molecules that can decrease fatty acid uptake and lipid synthesis/accumulation thus represent optimal candidates to reach this goal, either acting directly of the hepatocyte lipid transporter or on transcription factors able to regulate them. Although several molecules targeting different pathways have been developed and tested in animal models as well as in phase 2 and 3 clinical trials, none of them have been approved yet, either due to lack of efficacy or to the presence of side effects. It is thus possible that combination therapy will be necessary to tackle this health issue, allowing for a dosage reduction in single drugs in order to avoid adverse events and the discontinuation of therapy. Interesting data will also be provided by the ongoing trials, which include the use of molecules able to simultaneously tackle different proteins involved in lipid metabolism, such as PPARα and γ. Last but not least, more innovative approaches such as recombinant proteins or gene modifications through CRISPR-Cas9 will provide further tools for the managent of these disorders.

## Figures and Tables

**Figure 1 ijms-24-12748-f001:**
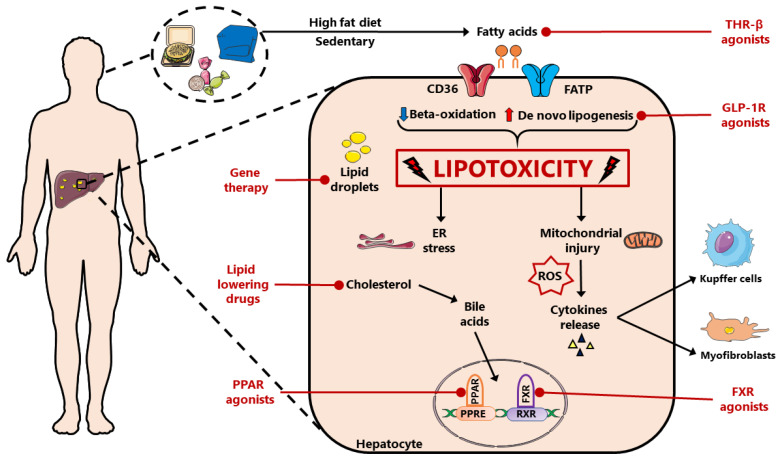
Schematic representation of NAFL/NASH pathogenesis and possible therapeutic approaches. Hepatic steatosis is the result of high diatary intake of fats and sugars together with a sedentary lifestyle. Ingested fatty acids are taken up from the circulation into the liver via CD36 and FATP transporters. Fatty acids are subsequently oxidized within the mitochondria by the β-oxidation process, stored within hepatic droplets or esterified into triglycerides to form VLDL. Insulin resistance, together with high fat consumption, can worsen the dysregulation of the lipid metabolism, leading to increased lipolysis. This kind of lipid impairment promotes oxidative stress, endoplasmic reticulum (ER) stress, and cytokines release, with the activation of Kupffer (KCs) and hepatic stellate cells (HSCs). As a consequence of the mitochondrial damage, reactive oxygen species (ROS) are produced, causing both lipid and protein peroxidation. Finally, the activation of HSCs favors the production of fibrillar collagen, leading to fibrotic scar deposition within the liver. In this complex setting, different therapeutic strategies have been developed in order to target the most significant pathways related to NAFL/NASH pathogenesis. Abbreviations: CD36: clustering domain 36; ER: endoplasmic reticulum; FATP: fatty acid transport protein; FXR: farnesoid X receptor; GLP-1: glucagon-like peptide 1; PPAR: peroxisome proliferator-activated receptor; PPRE: peroxisome proliferator responsive element; RXR: retinoid X receptor; ROS: reactive oxygen species; THR-β: thyrotropin-releasing hormone beta; VLDL: very low density lipoprotein. The image was created with the use of Servier Medical Art modified templates, licensed under a Creative Commons Attribution 3.0 Unported License (https://smart.servier.com), accessed on 1 June 2023.

**Figure 2 ijms-24-12748-f002:**
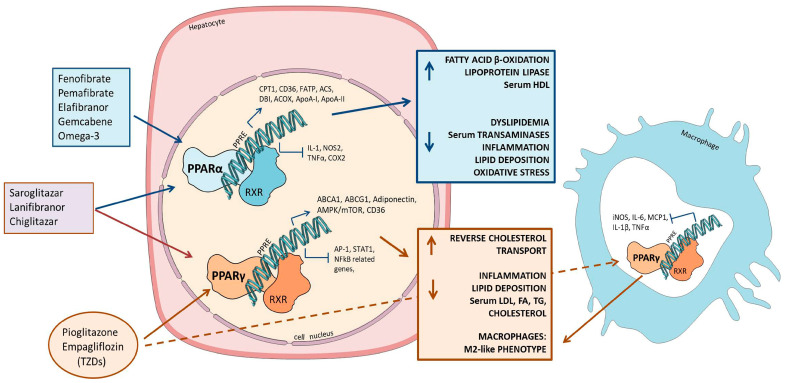
Schematic representation of PPAR targeting in NAFLD/NASH treatment. The main agonists of PPARα, PPARγ, or drugs acting on both are reported. PPAR agonists can directly induce or impede the mRNA synthesis of specific target genes, depending on the cell type. Here, the main molecular actions on hepatocytes and macrophages are reported, along with the main outcomes. Abbreviations: ABCA1: ATP-binding cassette transporter A1; ABCG1: ATP-binding cassette subfamily G member 1; ACS: acetyl-CoA synthetase; ACOX: acyl-CoA oxidase; AP-1: activator protein 1; ApoA-I: apolipoprotein A-I; ApoA-II: apolipoprotein A-II; COX2: cyclooxygenase 2; CPT1: carnitine palmitoyltransferase 1B; DBI: diazepam-binding inhibitor, acyl-CoA-binding protein; FATP: fatty acid transport protein; IL-1: interleukin 1; NOS2: nitric oxide synthase 2; PPAR: peroxisome proliferator-activated receptor; PPRE: peroxisome proliferator response element; RXR: retinoid X receptor, STAT1: signal transducer and activator of transcription 1; TNFα: tumor necrosis factor alpha; TZDs: thiazolidinediones. The image was created with the use of Servier Medical Art modified templates, licensed under a Creative Common Attribution 3.0 Unported License (https://smart.servier.com), accessed on 1 June 2023.

**Figure 3 ijms-24-12748-f003:**
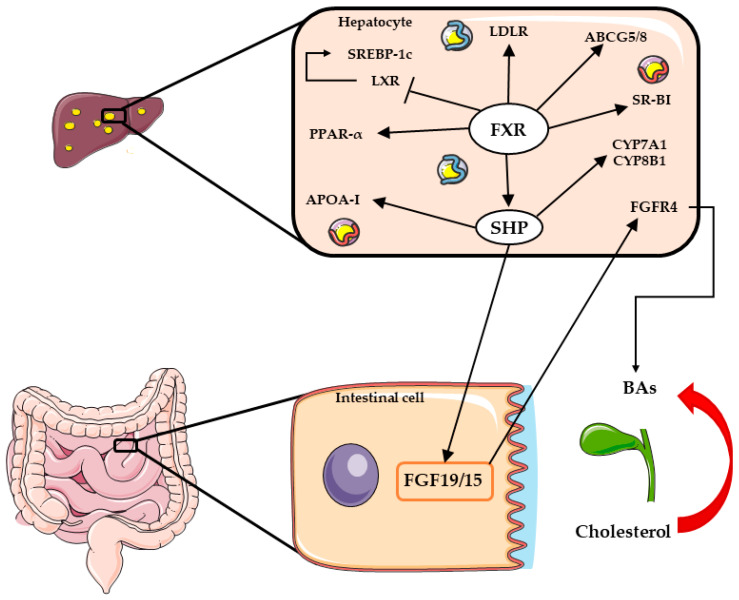
Schematic representation of FXR signaling within the liver. FXR signaling modulates both carbohydrate and lipid metabolism through the binding with SHP promoter. Due to the interaction with SHP, FXR inhibits the expression of NTCP (not shown), thus reducing the entrance of bile acids from the circulation. The modulation of LDLR by FXR is mediated by the inhibition of PCSK9 and the activation of reverse cholesterol transport (ABCG5/8). The activation of FXR also suppresses hepatic lipogenesis, decreasing the expression of SREBP1-c. In addition, upregulating PPARα, FXR also promotes free fatty acids β-oxidation and reduced VLDL production by the downregulation of MTP enzyme. Moreover, FXR acts at intestinal level, with the release of FGF19/15, which in turn bind to FGFR4 on the surfaces of hepatocytes, leading to the inhibition of CYP7A1 and CYP8B1. FXR activation is linked to the activation of SR-BI in the liver, reducing HDL plasma levels, consistent with the reduced synthesis of APO-AI. Abbreviations: ABCG5: ATP-binding cassette type 5; ABCG8: ATP-binding cassette type 8; APOA1: apolipoprotein A-I; APOB: apolipoprotein B; BAs: bile acids; CYP7A1: cytochrome P450 family 7 subfamily A member 1; CYP8B1: cytochrome P450 family 8 subfamily B member 1; FGF19: fibroblast growth factor 19; FGFR4: fibroblast growth factor receptor 4; FXR: farnesoid X receptor; GSK: glycogen synthase kinase; HDL: high-density lipoprotein; LDL: low-density lipoprotein; LDLR: low-density lipoprotein receptor; LXR: liver X receptor; MTP: microsomal triglyceride transfer protein; NTCP: Na^+^ taurocholate cotransporting polypeptide; PEPCK: phosphoenolpyruvate carboxy kinase; PCSK9: proprotein convertase subtilisin/kexin type 9; PPAR-α: peroxisome proliferator-activated receptor alpha; SHP: small heterodimer partner 1; SR-BI: scavenger receptor class B, type I; SREBP-1c: sterol regulatory element-binding protein 1; TG: triglycerides; VLDL: very low density lipoprotein. The image was created with the use of Servier Medical Art modified templates, licensed under a Creative Common Attribution 3.0 Unported License (https://smart.servier.com), accessed on 1 June 2023.

**Table 2 ijms-24-12748-t002:** List of PPAR-based drugs in trials for NAFLD/NASH.

Target	Identifier Trial	Condition	Drug	Phase	Status
pan-PPARs	NCT03459079	T2D and NAFLD	Lanifibranor	Phase 2	Recruiting
	NCT04849728	Fibrosis and NASH	Lanifibranor	Phase 3	Recruiting
	NCT05232071	T2M and NASH	Lanifibranor with SGLT2	Phase 2	Recruiting
	NCT03008070	NASH	Lanifibranor (IVA337)	Phase 2	Completed
	NCT05193916	NASH	Chiglitazar	Phase 2	Recruiting
PPAR α/γ agonists	NCT03639623	NASH	Saroglitazar	Phase 2	Recruiting
	NCT03617263	NAFLD + PCOS	Saroglitazar	Phase 2	Recruiting
	NCT05011305	Fibrosis and NASH	Saroglitazar	Phase 2	Recruiting
	NCT03061721	NAFLD + NASH	Saroglitazar	Phase 2	Completed
	NCT03863574	NASH	Saroglitazar	Phase 2	Completed
PPAR α/δ agonists	NCT01694849	NASH	Elafibranor (GFT505)	Phase 2	Completed
	NCT02704403 (RESOLVE-IT Trial)	NASH	Elafibranor	Phase 3	Terminated
	NCT03883607	NASH	Elafibranor	Phase 2	Terminated
	NCT03953456	NAFLD	Elafibranor	Phase 2	Terminated
PPAR δ agonist	NCT03551522	NASH	Seladelpar	Phase 2	Terminated
PPAR α agonists	NCT03350165	NAFLD	Pemafibrate	Phase 2	Completed
	NCT05327127	Fibrosis and NASH	Pemafibrate (K-877-ER) with SGLT inhibitor	Phase 2	Recruiting

The words “PPAR” AND “fatty liver disease”, “NAFLD”, “NASH” were searched on www.clinicaltrials.gov, accessed on 24 June 2023. PPAR: peroxisome proliferator-activated receptor.

**Table 3 ijms-24-12748-t003:** FXR-based drugs in trials for NAFL/NASH.

Target	Identifier Trial	Condition	Drug	Phase	Status
Nonsteroidal FXR agonists	NCT01999101	NAFLD	Cilofexor (Px-104)	Phase 2	Completed
	NCT02854605	NASH	Cilofexor (GS-9674)	Phase 2	Completed
	NCT02781584	NASH	Cilofexor with selonsertib and firsocostat	Phase 2	Completed
	NCT03449446	Cirrhosis (NASH)	Cilofexor with selonsertib and firsocostat	Phase 2	Completed
	NCT03987074	NASH	Cilofexor with semaglutide and firsocostat	Phase 2	Completed
	NCT03681457	NAFLD	Tropifexor (LJN452)	Phase 1	Completed
	NCT02855164	NASH	Tropifexor (LJN452)	Phase 2	Terminated
	NCT03517540	Fibrosis and NASH	Tropifexor with cenicriviroc	Phase 2	Completed
	NCT04065841	Fibrosis and NASH	Tropifexor with licogliflozin	Phase 2	Recruiting
	NCT03812029	NASH	Vonafexor (EYP001a)	Phase 2	Completed
	NCT03976687	Healthy and NASH	Vonafexor (EYP001a)	Phase 2	Completed
	NCT02913105	NASH	Nidufexor (LMB763)	Phase 2	Terminated
	NCT05397379	NASH	HEC96719	Phase 2	Recruiting
	NCT04328077	NASH	TERN-101	Phase 2	Completed
	NCT05338034	NASH	HPG1860	Phase 2	Active, not recruiting
	NCT04773964	NASH	MET642	Phase 2	Active, not recruiting
	NCT04702490	T2D and NASH	MET409	Phase 2	Active, not recruiting
Steroidal FXR agonists	NCT05573204	NASH	Obeticholic acid and vitamin E	Phase 2	Active, not recruiting
	NCT02548351	Fibrosis (NASH)	Obeticholic acid	Phase 3	Active, not recruiting
	NCT03836937	NAFLD	Obeticholic acid	Not applicable	Completed
	NCT01265498 (FLINT trial)	NASH	Obeticholic acid	Phase 2	Completed
	NCT02633956 (CONTROL trial)	Fibrosis (NASH)	Obeticholic acid	Phase 2	Completed
	NCT00501592	T2D (NAFLD)	Obeticholic acid (INT-747)	Phase 2	Completed
	NCT03439254 (REVERSE trial)	Cirrhosis (NASH)	Obeticholic acid	Phase 3	Completed
	NCT04378010	NASH	EDP-305	Phase 2	Recruiting
	NCT02918929	NAFLD and healthy	EDP-305	Phase 1	Completed
	NCT03421431	NASH	EDP-305	Phase 2	Completed

The words “FXR” AND “fatty liver disease”, “NAFLD”, “NASH” were searched on www.clinicaltrials.gov, accessed on 24 June 2023. FXR: farnesoid X receptor.

## Data Availability

Not applicable.

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
