# Peer review of "Current Therapeutical Approaches Targeting Lipid Metabolism in NAFLD"

_ijms, 2023, doi:10.3390/ijms241612748_

Round 1

Reviewer 1 Report

A review article by Vitulo et al. highlights the important role of lipid metabolism in the progression and development of NAFLD/NASH. The authors discuss the role of GLP, PPAR, and FXR, in NASH pathogenesis and also compiled several drugs under clinical trials. Overall, the manuscript provides updated information however, it has the following major concerns which need to be addressed:  

Abstract

This work gives a general look at the different therapeutic strategies currently being tested for NAFLD, other than or along with the recommendation of weight loss

In the abstract it seems that authors focused on NAFLD only however the title is for both NAFLD/NASH

 In introduction

Spelling error…..disfunctions

 I would recommend adding the Epidemiology of NAFLD/NASH

·         The global epidemiology of nonalcoholic fatty liver disease (NAFLD) and nonalcoholic steatohepatitis (NASH): a systematic review

·         Glucagon-like peptide 1 and fibroblast growth factor-21 in non-alcoholic steatohepatitis: An experimental to clinical perspective

·         The epidemiology of non-alcoholic steatohepatitis (NASH) in the United States between 2010-2020: a population-based study

Also discuss the Gut liver axis mechanism of NAFLD/NASH development

How lipid levels impact the Gut dysbiosis and its implication in pathogenesis.

Further, GLP is a incretin peptide and well correlated with the gut health hence also compile the relation between this

In figure 1 legends Abbreviation heading should be provided

Heading 1.1.

By the loss of triglycerides, VLDL particles are, through a complex series of reactions, converted in LDL particles which are triglyceride-  poor but cholesterol-rich.

Converted in change to converted into

PPARα activity is protective for liver steatosis, as demonstrated in animal models…..here sit should be protective against

In figure 2 & 3 legends Abbreviation heading should be provided

Table 1 is well compiled I would recommend to add dosage of drugs as well.

In Table 3 the trial no. NCT02654002 (Cilofexor) is compiled under FXR-based drugs in trials for NAFLD/NASH however this trial is not matching as the NCT02654002 (Cilofexor) trial is done for PK-PD study in healthy volunteers….( Study in Healthy Volunteers to Evaluate the Safety, Tolerability, Pharmacokinetics and Pharmacodynamics of GS-9674 (Cilofexor), and the Effect of Food on GS-9674 Pharmacokinetics and Pharmacodynamics)

Likewise there are other wrong information is provided hence before submitting the revised manuscript authors should check critically and revise the all the table accordingly

Please add challenges current perspective and challenges of use of Natural compounds

What is the current status of Natural compounds in clinical trials? Include the details.

Add one paragraph for obstacles/challenges in clinical trials

The conclusion needs to be improved with the involvement of lipid metabolism

Minor editing of English language required

Reviewer 2 Report

In the review article "Current therapeutical approaches targeting lipid metabolism in NAFLD/NASH" Vitulo et al., review key aspects of the potential therapeutical landscape of NAFLD. The review is timely and of great interest considering the high prevalence of NAFLD and its role in mixed dyslipidaemia. The review is well-written and grammatically impeccable. I however have different comments that should be addressed before considering the work acceptable for publication. Please find the main comments below.

Title: NAFLD includes NASH as it is a spectrum ranging from simple steatosis (NAFL) to NASH. Therefore, NASH is redundant. Please remove.

A clear definition of NAFLD and its main complications CVD, T2DM etc should be reported.

Overall, little attention has been paid to the last 20 years of literature regarding the lipidome beyond TG and cholesterol accumulation in NAFLD. Below some specific aspects to expand.

In the introduction it would be of help to the non-expert reader to highlight the different routes of hepatic fat accumulation (adipose tissue (59%), de novo lipogenesis (26%), diet (15%)). (PMID: 15864352).

When authors mention: “FFAs can be β-oxidized to produce energy, esterified to TGs used for the production of very low-density lipoproteins (VLDLs) [17] or stored in lipid droplets for later use” they should also mention that FA can be also esterified to other lipids and when in surplus (FAs) can go into lipotoxic lipid species such as diacylglycerols and ceramides (PMID: 34695228), which in turn promote the progression from NAFL to NASH.

When authors mention: “When carbohydrates are abundant, the liver converts glucose to FAs, a process called de novo lipogenesis (DNL)”, it should be specified that DNL is the production of lipids from non-lipid precursors (so not only glucose).

Authors mention “ABCA1 adds phospholipids to small HDLs, thus resulting in a pre-β HDL, whereas ABCG1 induces the addition of cholesterol to form larger HDL particles.” Both transporters can move PL and cholesterol as far as I am aware, so please provide adequate reference or amend.

NAFLD has a genetic component which also affects lipid metabolism. Is it known if the pharmacological treatment discussed have a different effect on patients with different genetic background?

“GLP-1 agonists have been approved for therapy in T2D (type 2 diabetes) and employed with discrete results to overcome insulin resistance, to decrease the appetite and favour weight loss, but they can also have direct effects on other tissues like the liver [64]”. It is now generally accepted that GLP-1 does not have a direct hepatic effect as there are no receptors in the liver (see work from Daniel Drucker or Fiona Gribble among many others). Moreover, a systematic review cannot be used as a reference to justify this statement. Provide original evidence or remove the line.

In the FFA section it should be mentioned that adipose tissue insulin resistance (inability of insulin to suppress lipolysis (PMID: 35192055)) is a crucial plyer in the onset and progression of NAFLD (PMID: 36378995, PMID: 22183689).

In the ApoA-I section it should be mentioned that HDL-C is also associated with NAFLD and that the lipidomic profile of the HDL particles is characterised by a depletion of PUFA phospolipids and enriched in ceramides (PMID: 37084865), features characterising the hepatic lipidomic profile of this condition (PMID: 29906416). This also align with the cholesterol efflux dysfunction discussed in the review.

Round 2

Reviewer 1 Report

Can be accepted for publication

Need minor corrections

Author Response

We do thank the reviewer, since with her/his comments has helped us to improve the overall quality of the manuscript.

We have further performed editing of English language.

Reviewer 2 Report

Authors have addressed my points. However, authors have not added the reference to some of the comments. Specifically, when they state "Systemic insulin resistance consists of impaired insulin-mediated suppression of hepatic glucose production. Adipose tissue IR refers to the loss of lipolysis suppression when insulin levels increasesd, and this is also associated with higher levels of circulating FFAs and increased hepatic gluconeogenesis". They should add the source (PMID: 35192055). 

When Authors mention "Serum analyses of individuals affected by NAFLD showed reduced levels of polyunsaturated phospholipids and FFAs, which were negatively correlated with the BMI, IR, TGs, and hepatocyte ballooning", they missed the reference to the original article (PMID: 37084865).

Author Response

As suggested, we have inserted the requested refences.